# Proline, Cysteine and Branched-Chain Amino Acids in Abiotic Stress Response of Land Plants and Microalgae

**DOI:** 10.3390/plants12193410

**Published:** 2023-09-28

**Authors:** Rachele Ingrisano, Edoardo Tosato, Paolo Trost, Libero Gurrieri, Francesca Sparla

**Affiliations:** Department of Pharmacy and Biotechnology FaBiT, University of Bologna, 40126 Bologna, Italy; rachele.ingrisano2@unibo.it (R.I.); edoardo.tosato2@unibo.it (E.T.); paolo.trost@unibo.it (P.T.); francesca.sparla@unibo.it (F.S.)

**Keywords:** amino acids, abiotic stress, proline, cysteine, hydrogen sulfide, catabolism, microalgae

## Abstract

Proteinogenic amino acids are the building blocks of protein, and plants synthesize all of them. In addition to their importance in plant growth and development, growing evidence underlines the central role played by amino acids and their derivatives in regulating several pathways involved in biotic and abiotic stress responses. In the present review, we illustrate (i) the role of amino acids as an energy source capable of replacing sugars as electron donors to the mitochondrial electron transport chain and (ii) the role of amino acids as precursors of osmolytes as well as (iii) precursors of secondary metabolites. Among the amino acids involved in drought stress response, proline and cysteine play a special role. Besides the large proline accumulation occurring in response to drought stress, proline can export reducing equivalents to sink tissues and organs, and the production of H_2_S deriving from the metabolism of cysteine can mediate post-translational modifications that target protein cysteines themselves. Although our general understanding of microalgae stress physiology is still fragmentary, a general overview of how unicellular photosynthetic organisms deal with salt stress is also provided because of the growing interest in microalgae in applied sciences.

## 1. Introduction

The role of amino acids in plant development and response to biotic and abiotic stress has received considerable attention in recent decades. Due to its potential applications, the study of amino acid metabolism and their role as sensing, signaling and protective molecules has attracted interest in the fields of basic and applied plant science.

As an example, by performing a quick search on Pubmed using the query “amino acid, plant, stress, metabolism, growth”, the number of publications retrieved is 5688 scientific papers published since 1973, of which 5455 are since only 2001. The topics covered in the present review, then, can be considered just a part of the multifaced role of amino acids in stress response. If the reader is interested in a wider perspective on amino acids, we suggest the following reviews [1,2,3,4,5].

Amino acids are carbon-based molecules, containing both an amino and a carboxyl functional group. One way to classify the proteinogenic amino acids synthesized by plants is based on their biosynthetic pathways [6]. Glucose derived from the photosynthetic fixation of CO_2_ can be considered as the source of carbon skeletons for the biosynthesis of amino acids, while the amino functional group primarily derives from nitrate or ammonia absorbed by the roots and fixed into glutamate to build up glutamine [7].

Tryptophan, phenylalanine and tyrosine are aromatic amino acids required for protein biosynthesis in all living cells. They all derive from chorismate, the end product of the shikimate pathway, and they play a crucial role in plants being precursors of a wide variety of bioactive molecules [8].

Cysteine, glycine and serine are all amino acids of the serine family. Serine is derived from the glycolytic intermediate 3-phosphoglycerate [9], which in turn can be converted into glycine by serine hydroxymethyl transferase [10]. Cysteine derives from serine and is the first organic molecule containing the reduced form of sulfate, i.e., the form of sulfur taken up by plants from the soil [11].

The amino acids of the aspartate family include aspartate, asparagine, lysine, threonine, methionine and isoleucine. This class is highly heterogeneous from a biochemical point of view [12]. Moreover, research studies focused on it because the absence of biosynthetic pathways for lysine, threonine, methionine and isoleucine in humans and animals makes some central enzyme of these pathways a potential target for new herbicides [13].

Alanine, leucine and valine are produced from pyruvate, the end product of glycolysis. Together with isoleucine, leucine and valine form the small group of branched-chain amino acids (BCAAs). Animals, including humans, are not able to synthesize BCAAs, thus they need to be absorbed through the diet. In plants, the breakdown of BCAAs is relevant in case of sugar deficiency, a condition often experienced by plants under drought conditions, to produce the energy necessary to survive stress [14].

Glutamate, glutamine, proline and arginine belong to the glutamate family. Glutamate has a central role in amino acid metabolism, being directly involved in both ammonia assimilation and transfer to all other amino acids [15]. Besides being building blocks in protein biosynthesis, proline and arginine contribute to stress tolerance in plants [16,17].

Finally, histidine is synthesized from ribose-5-phosphate through several reactions [18]. Histidine is characterized by an imidazole side group that allows the coordination of metal ions and acid–base catalysis.

Many recent and excellent reviews deal with the various physiological, metabolic and catabolic aspects connected to amino acids in plants. In the present one, we focus on the current knowledge on (i) how amino acids can efficiently replace sugars in the production of ATP and reducing power; (ii) the role of amino acids, e.g., arginine and lysine, as biosynthetic precursors for bioactive molecules; and (iii) the role of cysteine, the metabolism of which is connected to the physiological process of stomatal closure. A separate session is dedicated to microalgae, not because microalgae have different stress physiology than land plants, but rather because the study of microalgae is frequently driven by applied purposes.

## 2. Amino Acid Degradation Provides Energy for Enduring Abiotic Stress in Land Plants

Abiotic stresses such as drought, salinity, heat and cold affect plant physiology differently, but all impact photosynthesis. Drought and salinity interfere with stomata opening, leading to a decrease in the concentration of CO_2_ within the leaf and less carbon fixation. Heat stress affects transpiration and the stability of protein and membranes, resulting in lower carbon fixation. Cold stress slows down enzymatic reactions, resulting in an imbalance between the two phases of photosynthesis and reduced sugar production. Due to reduced CO_2_ uptake and fixation, plants can experience starvation under abiotic stress conditions. The decreased availability of newly formed carbon skeletons has a negative impact on mitochondrial energy production [19,20]. Plants address the issue with alternative carbon sources to sugars to endure and recover.

Amino acid oxidation can potentially provide similar amounts of reducing power and ATP as glucose oxidation [1], thereby constituting a valuable energy source. However, only a few amino acids are abundant in soluble form under normal growth conditions and the most abundant ones are glutamate and glutamine [21], two amino acids not inclined to fuel the mitochondrial transport chain. Regardless of this, protein degradation can occur under drought and other abiotic stresses, and the reduction of total protein content in response to drought suggests that amino acids otherwise locked inside proteins can be recycled for respiration [3,22].

Considering the Arabidopsis proteome, it has been estimated that conditions leading to a 10% increase in the most abundant free amino acids like glutamine can increase a less abundant amino acid like leucine by 3500% [23]. In this perspective, chloroplasts can be considered an important source of amino acids, as they contain the majority of cell proteins, mostly due to Rubisco. Stromal proteins can be degraded via Rubisco-containing bodies and transferred from chloroplasts to vacuoles [24]. Since Rubisco contains between 10 and 30% of leaf nitrogen in land plants [25], it could itself constitute a reservoir of amino acids under carbon/nitrogen shortage. In fact, under starvation, the change in concentration of basic, aromatic and branched chain amino acids (BCAAs; i.e., leucine, isoleucine and valine) in soluble form could partially reflect the amino acid composition of Rubisco [21].

The most energetic amino acids are tyrosine, leucine, lysine, isoleucine and proline [1]. In particular, BCAAs and lysine can supply electrons to the mitochondrial electron transport chain both indirectly, via conversion into tricarboxylic acids (TCA) cycle intermediates or acetylCoA [26], and directly, via the protein named electron transfer flavoprotein (ETF) that interacts with the ETF:ubiquinone oxidoreductase (ETF-QO) providing electrons to the electron transport chain [27,28] (Figure 1A). In accordance with their ability to provide reducing power to the electron transport chain, BCAA content increases from 3% to 23% under dark-induced carbon starvation [21].

The importance of protein autophagy under carbon starvation in the release of BCAAs and lysine is also indicated by the induction of the whole catabolic pathway of BCAAs and lysine that in *Arabidopsis thaliana* involves four specific enzymes: BCAA transaminase 2 (BCAAT), branched chain α-keto acid dehydrogenase complex (BCKDH), isovalerylCoA dehydrogenase (IVDH) and 2-hydroxyglutarate dehydrogenase, the last two of which serve as electron donors to the mitochondrial ubiquinol pool via ETF, also induced by carbon starvation [1,28,29,30,31,32,33,34] (Figure 1A). Nevertheless, it should be mentioned that BCAA de novo synthesis is induced under drought in different land plants [35,36].

## 3. Amino Acids as Precursors for Secondary Metabolites

Besides producing energy, amino acids can be consumed as precursors to produce useful compounds. In *A. thaliana* leaves, aromatic amino acids, serine and arginine decreased after salt or osmotic stress [37], and they are all involved in biosynthetic pathways.

Aromatic amino acids are precursors for the biosynthesis of alkaloids, glucosinolates (together with leucine, isoleucine, valine, alanine, methionine) and auxin, all biomolecules involved in defense, biotic and abiotic stress responses and growth [38,39].

Arginine is a precursor of polyamine biosynthesis; polyamines are accumulated under abiotic stress and play a role in protecting biomolecules and stimulating plant growth [40,41]. In Arabidopsis, overexpression of arginine decarboxylase 2 (*ADC2*), one of the key enzymes in polyamine biosynthesis, resulted in higher putrescine level and drought tolerance due to lower transpiration [42]. Conversely, silencing *ADC* genes negatively affected polyamine levels and led to growth defects [43]. Thus, the depletion of arginine observed during drought stress recovery might be related to polyamines [42,44,45].

Although primarily involved in the response to pathogens, the example of lysine is noteworthy because its degradation couples energy production with the synthesis of two compounds involved in Systemic Acquired Resistance (SAR). Indeed, lysine catabolism is induced by biotic and abiotic stresses [46,47,48]. Lysine degradation is connected to N-hydroxy-pipecolic acid and α-aminoadipic acid synthesis, two molecules involved in plant immunity and defense response [49,50]. The catabolic branch leading to aminoadipate involves the lysine ketoglutarate reductase (LKR)/saccharopine dehydrogenase (SDH) complex; after the reaction of an aldehyde dehydrogenase, α-aminoadipic acid is produced. This product can be used as a signal for SAR or further degraded to provide electrons to the mitochondrial electron transport chain [1,49]. The production of N-hydroxy-pipecolate occurs in a second catabolic branch involving the aminotransferase ALD1 and the reductase SARD4 together with other unknown reductases and the Flavin Monoxygenase 1 (FMO1) which finally catalyzes the N-oxidation of pipecolic acid to N-hydroxy-pipecolic acid [2,51]. Studies on Arabidopsis plants highlighted how the ALD1/FMO1 pathway is induced by pathogens while the LKR/SDH pathway is upregulated under salt or osmotic stress [49,51,52,53] (Figure 1A). Like α-aminoadipate, N-hydroxy-pipecolic acid is also involved in SAR signaling [54]. Lysine increases after *P. syringae* inoculation in Arabidopsis leaves together with its catabolites pipecolate and α-aminoadipic acid [55]. In agreement with this, treating plants with N-hydroxylated pipecolic acid is sufficient to induce systemic acquired response and pathogen resistance [56].

## 4. Proline Cycling for Osmotic Stress Response and Recovery

### 4.1. Proline Metabolic Pathway

Proline is one of the most studied amino acids involved in responses to drought, salinity and osmotic stress responses. Its molecular structure makes proline peculiar since the side chain is bound to the amino group of the main chain forming a five-atom ring. Proline is typically accumulated under water deficit and salt stress as a compatible osmolyte [57,58]. Proline has also shown a protective or stimulating effect on antioxidant activities [59] and scavenging properties against reactive oxygen species like H_2_O_2_, singlet oxygen and hydroxyl radicals [60,61,62].

Proline synthesis is activated by drought and salt stress, reaching up to millimolar levels in plant cells [59,63]. In leaves, proline accumulation is induced mostly at the transcriptional level by promoting the expression of the biosynthetic gene *PYRROLINE-5-CARBOXYLATE SYNTHASE* (*P5CS*), the first and key step of proline synthesis using glutamate, and consuming NADPH to produce pyrroline-5-carboxylate (P5C) (Figure 1B). Then, a second enzyme, P5C reductase (P5CR), converts P5C into proline with further consumption of NADPH. Proline synthesis is localized in the chloroplasts and cytoplasm [59], while proline catabolism occurs in mitochondria [4,64]. Two dehydrogenases are responsible for proline catabolism: proline dehydrogenase (ProDH) and pyrroline-5-carboxylate dehydrogenase (P5CDH). The first enzyme, ProDH, is considered the rate-limiting step of proline catabolism. Plant genomes typically contain two *ProDH* genes with different spatiotemporal expression and osmotic-stress-dependent regulation [65,66,67], characteristics that make ProDH the main controller of proline degradation. Under drought stress recovery, *ProDH* expression increases in leaves while *P5CS* expression is reduced, with the effect of decreasing proline levels and remobilizing carbon skeletons and reducing power [63,68,69,70].

### 4.2. Proline Provides Reducing Power to the Mitochondrial Electron Transport Chain

In mitochondria, proline catabolism generates reducing power from both ProDH and P5CDH, and these electrons can contribute to oxidative phosphorylation and, so, to ATP production [71] (Figure 1B). In Arabidopsis, ProDH has also been proposed to be tightly linked to the mitochondrial inner membrane, and so it might be able to transfer electrons directly to the ubiquinone pool in the electron transport chain [64]. Moreover, proline catabolism generates the TCA cycle intermediate oxoglutarate (OG) from glutamate, with further production of reducing power. Thus, accumulated proline can be considered as an energy store. Complete oxidation of proline can generate reductants that support the mitochondrial electron transport chain resulting in the formation of 25 ATP equivalents per proline molecule [1]. Although proline is not the only source of energy, it probably helps increase respiratory rates during recovery. The energetic function was shown to be exploited by exporting proline as a carrier of reducing equivalents into sink tissues and organs. Proline oxidation regenerates glutamate as well, allowing nitrogen remobilization from its derivatives glutamine and asparagine [71]. Both energy and nitrogen produced from proline degradation are apparently used under osmotic stress and may help during the recovery phase.

### 4.3. Proline Cycling Is Required for Drought Stress Tolerance in Arabidopsis thaliana

Although only proline synthesis was initially thought to be fundamental for drought tolerance, experimental evidence highlighted that both proline synthesis and catabolism are required for drought tolerance. In *p5cs1* and *prodh1* Arabidopsis mutants, the NADP^+^/NADPH ratio under drought stress is lower than in wild-type plants, suggesting that the cycling of proline from synthesis to degradation can contribute to regenerating NADP^+^ under these stress conditions [72,73,74]. NADP^+^ regeneration helps release the overexcitation from photosystems, decreasing the risk of cellular damage [19,20]. Drought stress is a clear example of proline cycling involving different tissues. In leaves, the synthesis contributes to regenerating NADP^+^ by consuming NADPH while, in roots, proline catabolism generates reductants (NADH and FADH_2_) to fuel drought stress responses through growth. In 2011, Sharma and colleagues [74] found that in the root apex the expression of *ProDH1* is induced under drought and high proline levels accumulate in the root apex of the *prodh1-2* line. These data, together with low oxygen consumption in *prodh1-2* roots and increased root growth under proline supplementation, suggest that proline accumulated in leaves is transferred to roots to sustain root growth under low water potential. Altogether, different experiments contributed to the hypothesis that proline cycling from synthesis to degradation, more than just proline accumulation, sustains osmotic stress responses and recovery in plants.

## 5. Cysteine and Cysteine Degradation in Drought Stress Response

### 5.1. Physico-Chemical Properties of Cysteine

Among the 20 proteinogenic amino acids, cysteine stands out for its peculiar physico-chemical features, specifically for the presence of the sulfur atom of the thiol group in its side chain that makes this amino acid highly reactive. The acid dissociation constant (p*K*_a_) of most cysteines, both in their free form and in their protein-constituting form, is generally quite high (>8); this means that in the plant, under physiological conditions, they are mostly in the protonated (or thiolic, -SH) state. However, some protein-bound cysteines are characterized by acidic p*K*_a_ values (ranging from 3 to 6.5), depending on the specific surrounding protein microenvironment; hence, they are mainly in the deprotonated (or thiolate, -S^−^) state [75,76]. Thiolate cysteine residues are more nucleophilic than thiolic ones; thus they can be more easily subjected to redox post-translational modifications (PTMs), which include the formation of inter- or intra-molecular disulfide bridges, reversible oxidation to sulfenic acid (-SOH), irreversible oxidation to sulfinic (-SO_2_H) or sulfonic (-SO_3_H) acid, S-glutathionylation (-SSG), S-nitrosylation (-SNO), S-cyanylation (-SCN) and S-persulfidation (-SSH). The physiological significance of these redox PTMs depends on the type of modification and the specific target protein: in general, cysteine-targeted redox PTMs can potentially affect the subcellular localization, catalytic activity, tridimensional structure or capability to interact with other proteins, but since there are no general rules to predict these effects, specific cases need to be investigated individually [75,76,77].

Besides being a component of proteins, as a free metabolite, cysteine is a precursor for the biosynthesis of several other molecules, including vitamins (e.g., thiamin, biotin), cofactors (e.g., coenzyme A, lipoic acid) and other amino acids (methionine), but it also represents a limiting factor in the biosynthesis of glutathione, one of the most important and abundant antioxidants in the plant cells (ranging from about 70–700 μM in the vacuole to 1.2–4.5 mM in plastids and cytosol) [78,79,80].

### 5.2. Cysteine Biosynthesis

Considering its redox reactivity, cysteine can become a toxic molecule for the organism when accumulated at high levels [81]. In order to maintain low intracellular concentrations of free cysteine (less than 10 μM in vacuoles and plastids, more than 300 μM in the cytosol [79]), biosynthesis needs to be finely balanced with its degradation [1].

The biosynthesis of cysteine takes place in mitochondria and chloroplasts but mostly in the cytosol, and consists of a two-step pathway: the first step is an acetylation reaction catalyzed by serine acetyltransferase (SAT), which generates O-acetylserine (OAS) from serine and acetylCoA, while the second reaction is catalyzed by OAS (thiol)lyase (OASTL), which produces cysteine by incorporating H_2_S (resulting from the reduction of the sulfate assimilated from the soil) into OAS. Therefore, the newly synthesized cysteine represents the first organic storage of the sulfur taken up from the environment [82] (Figure 2).

In plants and bacteria, SAT and OASTL can physically interact with each other and constitute the Cysteine Synthase Complex (CSC). The structural and physiological details of this mechanism have been reviewed in detail by Jez and Dey [83]: the CSC is formed thanks to the interaction between the C-terminal tail of SAT and the active site of OASTL; therefore, the formation of the complex catalytically inactivates the OASTL, while it was demonstrated that the SAT activity is enhanced. The association or dissociation of this bi-enzymatic complex is thought to be responsible for the modulation of the cysteine biosynthesis depending on the sulfur availability. Briefly, the proposed model is the following: when sulfur is available, SAT and OASTL form the complex, the activity of SAT is increased and the OAS is then converted into cysteine by the unbound, active OASTL; vice versa, under sulfur starvation conditions, OASTL cannot catalyze the incorporation of H_2_S into OAS, which accumulates and competes with the C-terminal region of SAT, thus determining the dissociation of the complex [83]. Consistently, the accumulation of OAS promotes the expression of genes involved in sulfate uptake [84]. In addition, it was also demonstrated that SATs are negatively regulated by allosteric feedback inhibition by cysteine itself [85].

### 5.3. Cysteine Degradation Occurs through Different Pathways

The degradation of cysteine mainly takes place in the mitochondria and cytosol. The most important cysteine catabolic pathway seems to be the one involving cytosolic desulfhydrases (DESs), which produce pyruvate, NH_3_ and H_2_S which, besides being essential for cysteine biosynthesis, is also a redox-signaling molecule with manifold cellular targets (Figure 2). Nevertheless, other degradation pathways exist in plants: cysteine desulfurases (also known as NifS-like proteins) are enzymes present in many subcellular compartments and they catalyze the transfer of sulfur from cysteine to iron–sulfur cluster scaffold proteins, thus producing alanine, that is in turn converted to pyruvate [1].

In mitochondria, the degradation of cysteine is tightly linked to the detoxification of cyanide (CN^−^) deriving mainly from ethylene biosynthesis. CN^−^ is a strong inhibitor of cytochrome oxidase (complex IV) of the electron transport chain. A β-cyanoalanine synthase (CAS) catalyzes the synthesis of β-cyanoalanine and H_2_S from CN^−^ and cysteine. β-cyanoalanine is then converted to either asparagine or aspartate and NH_3_ by a cytosolic nitrilase/nitrile reductase, while H_2_S needs to be detoxified, being another well-known inhibitor of the cytochrome oxidase; for this reason, it is used by the mitochondrial OASTL to synthesize cysteine. Therefore, this cyclic pathway of cysteine biosynthesis and degradation is important not only to ensure the proper functionality of the mitochondrial electron transport chain, but also to maintain appropriate CN^−^ levels, as this molecule has been shown to act as a signal for physiological processes such as seed germination and root development [82,86,87,88]. In agreement, *A. thaliana* mutants lacking either the mitochondrial CAS (CAS-C1 or CYS-C1) or the mitochondrial OASTL show a non-lethal accumulation of CN^–^ and H_2_S and are impaired in the correct development of root hairs [86,87].

Finally, another mitochondrial catabolic pathway was identified in *A. thaliana*: according to the model described in [89], cysteine is first converted to 3-mercaptopyruvate by a yet-unidentified enzyme and then the sulfur is transferred to glutathione, probably by sulfur transferase 1, thus producing pyruvate and S-sulfanylglutathione (GSSH), which is a substrate for ethylmalonic encephalopathy protein1 (ETHE1) [1,89]. ETHE1 is a sulfur dioxygenase which eventually oxidizes GSSH, producing sulfite (SO_3_^2−^) and regenerating glutathione. Interestingly, *ethe1-1* mutants show phenotypic traits similar to those of plants lacking components of the systems that oxidize BCAAs as alternative energy sources, suggesting the involvement of ETHE1 in both processes [89].

### 5.4. The Role of Cysteine Metabolism in Drought Stress Response

In recent years, the connection between the drought stress response and H_2_S-mediated redox signaling has been extensively investigated in *A. thaliana*. Together with other signals such as strigolactones, small peptides and hydraulic stimuli [90], sulfate (SO_4_^2−^) was demonstrated as acting as a long-distance messenger of drought stress conditions. When soil water availability is limited, SO_4_^2−^ is transported by the xylem from the roots to the leaves, where it is incorporated into cysteine (Figure 2). The resulting increase in intracellular cysteine promotes the expression of the *NINE-CIS-EPOXYCAROTENOID DIOXYGENASE 3* (*NCED3*) gene. The enzyme NCED3 is considered the rate-limiting enzyme in the abscisic acid (ABA) biosynthesis. Concomitantly, cysteine acts as the substrate of the molybdenum cofactor sulfurase ABA3, which is needed for the activation by PTM of the last enzyme of the ABA biosynthetic pathway (abscisic aldehyde oxidase, AAO3M; Figure 2). Thus, the accumulation of cytosolic cysteine correlates with the increase in ABA, an event particularly relevant in guard cells [91].

Several members of the canonical ABA signaling pathway are essential for the sulfate-mediated stomatal closure, among which are the phosphatase ABA insensitive 1 (ABI1), the central kinase open stomata 1 (OST1, also known as SnRK2.6) and the NADPH oxidases respiratory burst oxidase homologue D and F (RBOHD and RBOHF) (Figure 2). Arabidopsis plants mutated in each of these loci are in fact impaired in the stomatal closure when treated with exogenous sulfate [92].

In guard cells, the increase in ABA content promotes the expression of *DES1* encoding for the cytosolic enzyme DES which catalyzes the degradation of cysteine to pyruvate, NH_3_ and H_2_S [3,93,94,95]. Interestingly, as the ABA biosynthesis in guard cells is essential for the expression of *DES1*, DES1 itself was also shown to be necessary for the expression of several ABA biosynthetic genes, such as *ZEAXANTHIN DEEPOXYDASE (ZEP), NCED3, AAO3* and *ABA3* [96]. The accumulation of H_2_S produced by the DES1-mediated cysteine catabolism determines the S-persulfidation of a number of target proteins involved in the closure of stomata [97]. The targets identified so far include (i) ABA insensitive 4 (ABI4), a transcription factor which is activated and promotes the expression of *DES1* itself and *MAPKKK18*, a member of a signaling cascade involved in drought stress responses [98,99] (Figure 2); and (ii) OST1, the persulfidation of which increases its phosphorylating activity and mediates stomatal closure through the activation of both RBOHF and slow anion channel 1 (SLAC1) (Figure 2). In addition, persulfidated OST1 promotes its interaction with ABA response element-binding factor 2 (ABF2), a transcription factor acting downstream in the ABA signaling, and enhances the ABA-induced cytosolic Ca^2+^ signaling, overall promoting the closure of stomata [100,101,102]. Interestingly, among the H_2_S targets, there is also (iii) DES1 which is subject to a positive feedback regulation, since the transient accumulation of H_2_S persulfidates two cysteine residues of DES1 itself [103]. Finally, (iv) the NADPH oxidase RBOHD is also activated by S-persulfidation, causing an ROS burst that amplifies the ABA signal stimulating stomatal closure and, at the same time, switches off the circuit acting as negative regulator of DES1 in competition with the persulfidation [103]. In addition, H_2_S was also demonstrated to selectively inhibit inward-rectifying K^+^ channels in tobacco guard cells, thus contributing further to stomatal closure [104].

In the stomata closure pathway, cross talk between H_2_S and nitric oxide (NO) has been also suggested, with DES1-produced H_2_S acting upstream of NO, presumably modulating the activity of nitrate reductase (NR) [94].

This whole model was very recently corroborated by Jurado-Flores et al. [105]: the authors showed that the pre-treatment of Arabidopsis plants with H_2_S donors significantly diminishes the levels of several drought biomarkers (such as proline, H_2_O_2_ and anthocyanins), but they also highlighted the H_2_S-mediated protein persulfidation as a crucial molecular defense mechanism against drought stress.

## 6. Microalgae: Adaptation Strategies to Salinity Stress Include a Central Role for Amino Acids

A generally accepted definition of microalgae includes an informal group of unicellular photosynthetic organisms encompassing prokaryotic (e.g., cyanobacteria) and eukaryotic (from the model green algae *Chlamydomonas reinhardtii* to evolutionary enigmatic diatoms) organisms. Constituting such a broad class of microorganisms (mainly grouped as a function of their size; 3–10 µm), microalgae occupy a wide range of environments and, as primary producers, provide the base of the aquatic ecosystem trophic chain [106,107].

In the last decade, microalgae have received increasing interest primarily as biofactories for feedstock in biofuel production and as a source of high added-value molecules [108,109]. As a consequence, applied research on microalgae is advancing rapidly, while still little is known about their physiology and mechanisms of adaptation to environmental stress. In particular, the knowledge of the role of amino acids in response to stress conditions is also still fragmentary because the specific profiles and levels of amino acids can vary according to the species of microalgae and the type, duration and intensity of stress [110].

Salinity is one of the most significant stressors for microalgae [111]. Global salinity profiles, which are constantly changing due to climate change, influence the distribution of microalgal communities [112]. High salinity represents a complex challenge for photosynthetic organisms because of ionic and osmotic impairments to the ability to take up nutrients and maintaining water balance. These phenomena impact cellular electron chains, mostly photosynthesis, generating reactive species and so oxidative stress [113]. To cope with dehydration and restore the intracellular osmotic balance, land plants accumulate compatible solutes such as betaines, proline and sugar alcohols [37,114]. Amino acids play an important role as osmoregulators also in microalgal cells, in which they are the most significant metabolite group among the dysregulated compounds in response to salt stress [115].

Microalgae exhibit a greater repertoire of adaptive responses than plants, presumably due to their unicellular organization. Consequently, an overall picture of the stress response is difficult to draw. Among the most studied microalgae, *Chlamydomonas* spp. overcomes unfavorable conditions by acquiring the osmoprotective morphology known as a “palmelloid” state [112,116]. Even the hypersalinity-adapted strains of *Dunaliella salina*, which are surrounded by an elastic plasma membrane and lack a rigid cell wall, cope with stress mainly by increasing their cell size to restore turgor pressure and producing glycerol as an osmoprotectant [117]. In contrast, in the *Chlorella* spp., which has a rigid cell wall and lacks the ability to change its cells morphology, the main osmolytes produced are the amino acid alanine, glutamic acid and asparagine but mostly proline, glycine and glycine-betaine [118]. Glycine-betaine is an amino-acid-derived osmolyte known to stabilize the integrity of cell membranes from the detrimental effect of high salt [118]. Proline is the most common compatible solute in land plants, where it also acts as a non-enzymatic ROS scavenger [119]. Microalgae cells (e.g., *Acutodesmus dimorphus*, *Chlamydomonas reinhardtii*, *Nostoc ellipsosporum*, *Scenedesmys* sp. CCNM 1077) subjected to different environmental stress conditions show a concomitant increase in oxidative stress resulting from the accumulation of ROS and proline, suggesting a common stress-adaptation strategy between terrestrial and aquatic organisms [120,121,122,123].

Although proline is often used as a stress indicator, recent evidence has shown that it is not the only one. A study on the adaptation mechanisms of freshwater *Chlorella* spp. to 30 g/L NaCl by exploiting the Adaptative Laboratory Evolution technique showed that, after 138 days of treatment, genes involved in the metabolism of tryptophan, histidine, glycine, serine and threonine were up-regulated, while those of tyrosine were down-regulated [124]. An extensive comparative metabolomic study about the different osmoadaptations to hypersalinity of three diatoms species demonstrated that certain amino acids are specific for short-term or long-term adaptation [115]. For example, in *Skeletonema marinoi*, a dysregulation of leucine content was detected, specifically an accumulation during short-term adaptation and a decrease under long-term response. The authors advanced the hypothesis that, at first, leucine is accumulated due to a reduced protein synthesis rate and then it is degraded to provide an alternative source of acetylCoA, maintaining the overall metabolism [115].

## 7. Concluding Remarks

Climate change increases the severity of damages to plants and significantly impacts crop productivity. Within plants, amino acids perform various functions and some of them are strictly related to drought stress. Considerable progress has been made in understanding the physiological framework within which proline and cysteine act. Similarly, the role of BCAAs as electron donors to the mitochondrial electron transport chain is well understood. Due to this knowledge, the use of environmentally friendly biostimulants based on or containing amino acids is becoming a common practice. However, while the role of BCAAs has been studied under carbon starvation, it would be important to understand if they have a role in physiological plant growth, for example compensating energy demand under a short-term lack of carbon or energy. The mechanism involving cysteine in signaling for stomata movement has been detailed in many aspects, but a more general comprehension of cysteine-derived signaling has yet to be obtained. For both cases, translational works to understand the function of BCAA and cysteine in crops are still lacking.

Unicellular microalgae occupy an important niche as biofactories and we consume microalgae as food, feed and starting material for the extraction of high-added-value products. Microalgae-based applications are continuously increasing, often ignoring that understanding the physiology of these organisms would tremendously increase our ability to utilize them in future applications. For this reason, our knowledge of microalgae stress physiology requires further exploration, including the role of amino acids and beyond.

## Figures and Tables

**Figure 1 plants-12-03410-f001:**
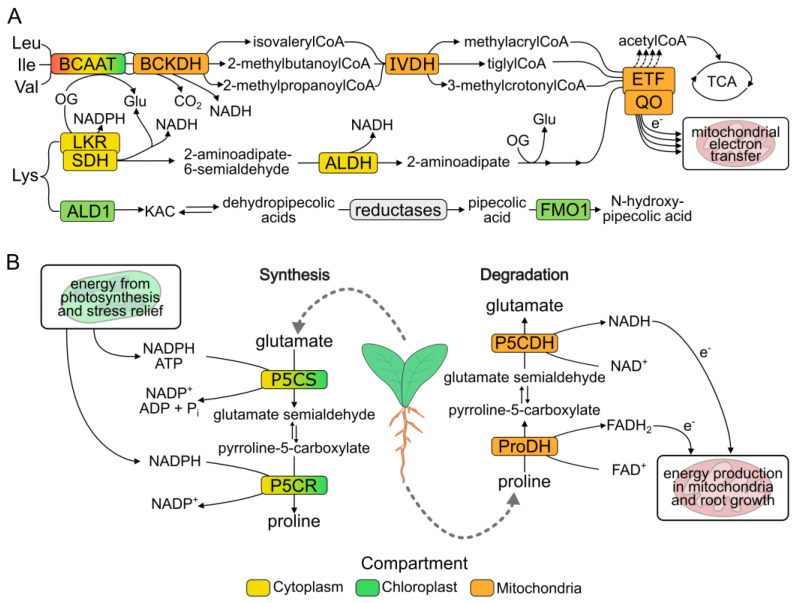
Metabolism of amino acids under stress conditions. (**A**), catabolic reactions involving BCAA or lysine producing electrons, supplied to oxidative phosphorylation in mitochondria and acetylCoA for tricarboxylic acid cycle (TCA). Catabolism of lysine provides aminoadipate and N-hydroxy-pipecolic acid, signal molecules for SAR. Reductases involved in dehydropipecolic acids reduction are not completely identified; SARD4 has been discovered but it is not the unique enzyme catalyzing this conversion. ALD1, aminotransferase; ALDH, aldehyde dehydrogenase; BCAAT, BCAA transaminase; BCKDH, branched chain α-keto acid dehydrogenase; ETF, electron transfer flavoprotein; ETF-QO, electron transfer flavoprotein:ubiquinone oxidase; FMO1, flavin monoxygenase 1; IVDH, isovalerylCoA dehydrogenase; LKR, lysine-ketoglutarate reductase; SDH, saccharopine dehydrogenase. (**B**), proline cycling between synthesis and turnover. In leaves protein is synthetized, especially consuming reducing power from photosynthesis. Proline is putatively transferred in roots, where its catabolism provides electrons to fuel the mitochondria electron chain and then root growth. ProDH, proline dehydrogenase; P5CDH, pyrroline-5-carboxylate dehydrogenase; P5CR, pyrroline-5-carboxylate reductase; P5CS, pyrroline-5-carboxylate synthase. Enzyme localization is shown as background color and explained in the legend.

**Figure 2 plants-12-03410-f002:**
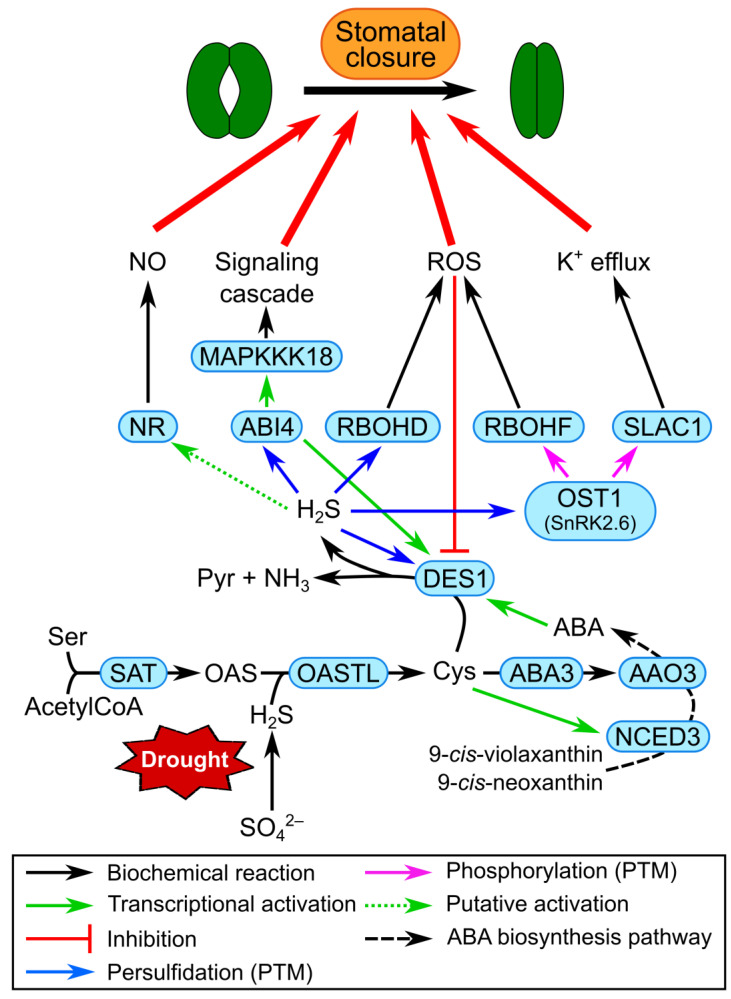
Schematic representation of cysteine biosynthesis and signaling pathways that lead to stomatal closure from cysteine catabolism under drought stress conditions. Cytosolic DES1-mediated cysteine degradation determines the accumulation of H_2_S, which is responsible for the S-persulfidation and consequent activation of several target proteins, giving rise to different cellular responses that converge on the closure of stomata. AAO3, abscisic aldehyde oxidase3; ABA, abscisic acid; ABI4, ABA insensitive4; Cys, cysteine; DES1, L-cysteine desulfhydrase1; MAPKKK18, mitogen-activated protein kinase kinase kinase18; NCED3, nine-cis-epoxycarotenoid dioxygenase3; NR, nitrate reductase; OAS, O-acetlyserine; OASTL, OAS (thiol)lyase; OST1, open stomata1; PTM, post-translational modification; Pyr, pyruvate; RBOHD/F, respiratory burst oxidase homologue D/F; ROS, reactive oxygen species; SAT, serine acetyltransferase; Ser, serine; SLAC1, slow anion channel1.

## Data Availability

No new data were created or analyzed in this study. Data sharing is not applicable to this article.

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
