# Peer review of "Proline, Cysteine and Branched-Chain Amino Acids in Abiotic Stress Response of Land Plants and Microalgae"

_plants, 2023, doi:10.3390/plants12193410_

Round 1

Reviewer 1 Report

The manuscript received for review is an example of a review paper that organizes contemporary information on the participation of amino acids in plant response to stress, especially abiotic stress, in a competent and interesting way. The authors point to the role of these substances as a source of metabolic energy and precursors in the biosynthesis of secondary metabolites. Attention is also drawn to the special role of proline and cysteine and to the specificity of stress response in microalgae. No significant factual errors were found in the manuscript. However, it seems to me that there is some inconsistency in the structure of the division of the paper. In the title of chapter 5, it was emphasized that the presented content concerns microalgae. However, in the titles of earlier chapters there is no reference to the plants they refer to. This may suggest that these parts present general content (possibly also concerning microalgae), while chapter 5 repeats the information concerning only this group of plants. In addition, the figur captions contain information on the mechanisms of the considered reactions (lines 96-101, fig. 1 and lines 258-261, fig. 2), which in my opinion should be included in the main text of the work.

Author Response

The manuscript received for review is an example of a review paper that organizes contemporary information on the participation of amino acids in plant response to stress, especially abiotic stress, in a competent and interesting way. The authors point to the role of these substances as a source of metabolic energy and precursors in the biosynthesis of secondary metabolites. Attention is also drawn to the special role of proline and cysteine and to the specificity of stress response in microalgae. No significant factual errors were found in the manuscript.

> However, it seems to me that there is some inconsistency in the structure of the division of the paper. In the title of chapter 5, it was emphasized that the presented content concerns microalgae. However, in the titles of earlier chapters there is no reference to the plants they refer to. This may suggest that these parts present general content (possibly also concerning microalgae), while chapter 5 repeats the information concerning only this group of plants.

A: We thank the Reviewer for the useful comments. In the revised version of the manuscript the distinction between the organisms discussed in the various sections has been improved, by specifying in the section titles and within the text which organisms we are referring to.
See lines: 80; 94; 198; 212; 215; 334.

>In addition, the figure captions contain information on the mechanisms of the considered reactions (lines 96-101, fig. 1 and lines 258-261, fig. 2), which in my opinion should be included in the main text of the work.

A: We thank again the Reviewer for the comment, the content of captions of Figure 1 and Figure 2 can be found in the text. See lines 136-138 and 143-146 for the caption of Figure 1 and line 353-365 for the caption of Figure 2.

Reviewer 2 Report

The article has a promising title, but in fact contains little content regarding the role of plant amino acids in the stress response. Such information is very abundant in the literature. So, for the query ‘amino acids AND stress AND plants’ (https://scholar.google.com/scholar?start=10&q=amino+acids+AND+stress+AND+plants&hl=ru&as_sdt=0.5) found more than 1.5 million articles and books, where various aspects of the participation of amino acids in plant stress responses are described and discussed in detail. The authors do not mention the numerous studies and reviews previously published in this area, neither in the Introduction nor in the body of the review. At the same time, all the main conceptual ideas written in the peer-reviewed article have already been published and discussed in detail earlier.

The review is poorly structured and does not answer the questions posed in the title. Most of manuscript is occupied by the consideration of metabolic pathways for the degradation of amino acids and the subsequent synthesis of new biomolecules. Actually, there is very little information about stresses in the article. Only sections 3.3 and 4.4 contain very little information on the involvement of proline and cysteine, respectively, in the response to drought stress. There are no data on the participation of amino acids in response to other stresses.

Section 5 about microalgae looks very strange. This section looks like a foreign body in the article, since all the previous material was on the example of higher plants. Section 5.1 has nothing to do with the research topic. Section 5.2 contains very little data on the involvement of amino acids in the response to salinity stress. In section 5.3 there is no information about stresses, and almost nothing about amino acids.

Obviously, the authors should radically restructure the article, dividing it into several articles that locally shown current knowledge in the areas of: (i) amino acid metabolism; (ii) the contribution of specific amino acids to the response of plants to specific abiotic and/or biotic stresses; (ii) amino acids in microalgae and their response to stress.

The manuscript as presented is not suitable for publication.

Author Response

>The article has a promising title, but in fact contains little content regarding the role of plant amino acids in the stress response. Such information is very abundant in the literature. So, for the query ‘amino acids AND stress AND plants’ (https://scholar.google.com/scholar?start=10&q=amino+acids+AND+stress+AND+plants&hl=ru&as_sdt=0.5) found more than 1.5 million articles and books, where various aspects of the participation of amino acids in plant stress responses are described and discussed in detail. The authors do not mention the numerous studies and reviews previously published in this area, neither in the Introduction nor in the body of the review. At the same time, all the main conceptual ideas written in the peer-reviewed article have already been published and discussed in detail earlier. The review is poorly structured and does not answer the questions posed in the title. Most of manuscript is occupied by the consideration of metabolic pathways for the degradation of amino acids and the subsequent synthesis of new biomolecules. Actually, there is very little information about stresses in the article. Only sections 3.3 and 4.4 contain very little information on the involvement of proline and cysteine, respectively, in the response to drought stress. There are no data on the participation of amino acids in response to other stresses.

A: We thank the reviewer for the stimulating critics, and we are sorry he/she does not consider our work worth to be published. Amino acids in plant stress is a wide topic touching many processes and pathways. Considering this, we chose to focus on a few stresses, mainly drought and salt stress, also for their relevance on a global perspective. Summarizing in a short review all the discoveries involving amino acids was far from our intentions. Our idea was to focus on phenomena related to stress and amino acids that emerged more clearly in the last years, i.e., amino acids degradation under carbon starvation, proline production to cope with stress and reducing power transfer from shoot to root, cysteine as a source of signals related to stress.
Despite the number of available publications responding to the query “amino acids”, it must be considered that many studies mention amino acids without giving any molecular clue, just reporting changes in levels, expression of related genes, testing them as a treatment. Using most of this information would lead to lists of amino acids involved in some stress more than explanations of mechanisms, which indeed was our intention.
The pathways mentioned in the manuscript are induced and stimulated by stresses, for this reason they were explained. A clear example is the degradation of BCAA to provide energy under carbon starvation. Carbon starvation is induced by many stresses, drought among all, by affecting photosynthesis and so carbon availability.

We recognize that the title of the manuscript was too general, and a more specific one was needed.  We modified the title of the manuscript to make clearer the focus of the review (now “Amino acids under stress: focus on energy and signaling in the stress response of land plants and microalgae”), an introductive section was also added providing first general information on amino acids and then introducing the focus of the manuscript. In the Introduction section a sentence (line 35-36) has been included to provide the reader with suggested reviews, giving the idea of the variety and complexity of amino acid metabolism under stress.

>Section 5 about microalgae looks very strange. This section looks like a foreign body in the article, since all the previous material was on the example of higher plants. Section 5.1 has nothing to do with the research topic. Section 5.2 contains very little data on the involvement of amino acids in the response to salinity stress. In section 5.3 there is no information about stresses, and almost nothing about amino acids.

A: The role of amino acids in microalgae metabolism is really understudied. Our goal was to summarize the few information from reliable studies on amino acids and provide an initial idea on the topic. We had dived deeply into the literature about microalgae, stress and amino acids and it was very hard to find useful information. Anyway, considering the increasing importance of microalgae in research and economy, we thought it was worth to provide information on the topic to start building a reference on it.
We recognize that section 5.1 is quite off topic, not talking about amino acids but only about stress, and section 5.3 poorly regards stress, even if mentions the interest on BCAA to modify microalgae metabolism. The whole section on microalgae has been modified to make the content more consistent in the perspective of the review, both text and titles have been improved. Section 5.1 (now 6.1) has been deeply modified and joined with section 5.2, while we remodeled section 5.3 (now 6.2).

Obviously, the authors should radically restructure the article, dividing it into several articles that locally shown current knowledge in the areas of: (i) amino acid metabolism; (ii) the contribution of specific amino acids to the response of plants to specific abiotic and/or biotic stresses; (ii) amino acids in microalgae and their response to stress.

The manuscript as presented is not suitable for publication.

Reviewer 3 Report

The authors focus on amino acids that contribute to the plant stress response. The work is interesting. I have several important points that require attention from the authors.

1.       Please add an introduction section with more points regarding the aims and significance of this study.

2.       Overall classification, and properties of amino acids are good to add those are responsible for stress response.

3.       Figures 1 and 2 look good. For Figure 2, I suggest adding stomatal closure images to enhance its visual appeal.

4.       I don’t find any research gap, future prospects in this study well. It can be described in the new section.

5.       Please update the recent references throughout the manuscript.

Author Response

The authors focus on amino acids that contribute to the plant stress response. The work is interesting. I have several important points that require attention from the authors.

  1. Please add an introduction section with more points regarding the aims and significance of this study.

A: We thank the Reviewer for the positive opinion and the useful comments. We included an introduction section in which we summarize the main characteristics of the proteinogenic amino acids and explain the organization and aim of the review.

  1. Overall classification, and properties of amino acids are good to add those are responsible for stress response.

A: Thanks for the comment and, as suggested, we have exposed the properties of amino acid in the newly added Introduction section. Furthermore, in the section 4 on proline and section 5 on cysteine more properties, also related to stress, have been reported.
See lines: 174-178; 234-246

  1. Figures 1 and 2 look good. For Figure 2, I suggest adding stomatal closure images to enhance its visual appeal.

A: A schematic representation of stomata has been included in Figure 2, thanks for the suggestion.

  1. I don’t find any research gap, future prospects in this study well. It can be described in the new section.

A: In the concluding remarks (now section 7) the future perspectives have been improved and expanded, changes are highlighted in red.

  1. Please update the recent references throughout the manuscript.

A: For this purpose, we reanalyzed the literature and found other useful references to add. Furthermore, other references were included thanks to the new Introduction section. All the new citations, 26 in total, are highlighted in red in the References sections.

Round 2

Reviewer 2 Report

The manuscript has been revised and improved to some extent, mainly in changing the title and introduction. However, I still have objections to publishing the article in this, even modified, presentation.

Firstly, the vast majority of sections contain abstract discussions about stress, despite the fact that all plant stresses are very different in their physiological and biochemical essence. Therefore, when describing the reaction of plants to stress, it is necessary to analyze the stress factor itself and only then the reaction to it.

Secondly, the article analyzes only osmotic stresses - the effect of drought (on terrestrial plants) and salinity (on microalgae). In this regard, the authors need to make changes to the title of the article and more clearly state the matters studied and discussed in the article.

Thirdly, describing the role of proline only and under drought stress only does not meet the objectives of the article. It is necessary to indicate the effect of stress on other amino acids, or change the title of the article.

Fourthly, section 6.2 (about changing the metabolism of microalgae in the direction of increasing the synthesis of triacylglycerols) is not relevant to the topic of the manuscript.

Overall, my opinion of the manuscript remains the same. It needs to be finalized/redesigned, making it more structurally clear and rich in specific material that responds to the main theme presented in the title.

Author Response

The manuscript has been revised and improved to some extent, mainly in changing the title and introduction. However, I still have objections to publishing the article in this, even modified, presentation.

Firstly, the vast majority of sections contain abstract discussions about stress, despite the fact that all plant stresses are very different in their physiological and biochemical essence. Therefore, when describing the reaction of plants to stress, it is necessary to analyze the stress factor itself and only then the reaction to it.

  • Stress conditions have been made explicit substituting “stress” with the kind of stress reported in references. A brief description of the effect of abiotic stresses on plants have been added in section 2 and section 6.

Secondly, the article analyzes only osmotic stresses - the effect of drought (on terrestrial plants) and salinity (on microalgae). In this regard, the authors need to make changes to the title of the article and more clearly state the matters studied and discussed in the article.

  • We recognize that the title was still too general to reflect the actual content of the review, for this reason the title has been changed to make it more coherent with the contents.

Thirdly, describing the role of proline only and under drought stress only does not meet the objectives of the article. It is necessary to indicate the effect of stress on other amino acids, or change the title of the article.

  • With this review we wanted to describe the few molecular mechanisms that are more characterized involving amino acids and photosynthetic organisms. It was far from our idea to treat all the amino acids in all the stresses that are reported, also because for most of them detailed information is still missing and just eventual changes in their levels are known. Proline, as cysteine, have been treated in the perspective of drought because it is under drought that the mechanisms described have been highlighted. However, we agree that in the current form the title is misleading and coherently with our aim we changed the title to make clear and explicit the contents of the manuscript.

Fourthly, section 6.2 (about changing the metabolism of microalgae in the direction of increasing the synthesis of triacylglycerols) is not relevant to the topic of the manuscript.

  • We removed the section.

Overall, my opinion of the manuscript remains the same. It needs to be finalized/redesigned, making it more structurally clear and rich in specific material that responds to the main theme presented in the title.

  • We sincerely regret the still negative judgment.